# Technologies in Biomarker Discovery for Animal Diseases: Mechanisms, Classification, and Diagnostic Applications

**DOI:** 10.3390/ani15213132

**Published:** 2025-10-29

**Authors:** Salwa Eman, Raza Mohai Ud Din, Muhammad Hammad Zafar, Mengke Zhang, Xin Wen, Jiayu Ma, Ahmed A. Saleh, Hosameldeen Mohamed Husien, Mengzhi Wang, Xiaodong Guo

**Affiliations:** 1Collage of Animal Science and Technology, Yangzhou University, Yangzhou 225009, China; mh24126@stu.yzu.edu.cn (S.E.); mh24136@stu.yzu.edu.cn (R.M.U.D.); hammadzafar075@gmail.com (M.H.Z.); mz120231610@stu.yzu.edu.cn (X.W.); 19549536595@163.com (J.M.); elemlak1339@gmail.com (A.A.S.); 008643@yzu.edu.cn (H.M.H.); mengzhiwangyz@126.com (M.W.); 2College of Food Science and Light Industry, Nanjing Tech University, Nanjing 211816, China; zhangmengke0725@163.com; 3Animal and Fish Production Department, Faculty of Agriculture (Al-Shatby), Alexandria University, Alexandria City 11865, Egypt; 4College of Veterinary Medicine, Albutana University, Rufaa 22217, Sudan; 5State Key-Laboratory of Sheep Genetic Improvement and Healthy-Production, Xinjiang Academy of Agricultural Reclamation Sciences, Shihezi 832000, China; 6Joint International Research Laboratory of Agriculture & Agri-Product Safety of MOE, Yangzhou University, Yangzhou 225009, China

**Keywords:** animal disease, biomarkers, early diagnosis, technologies

## Abstract

The early detection of animal diseases is challenging because clinical signs often appear late or are non-specific. This review explores how biomarkers, measurable molecular indicators, within an animal’s body can assist veterinarians in identifying diseases earlier, predicting their progression, and monitoring treatment response. It highlights the effectiveness of advanced laboratory techniques (including genomics, proteomics, and metabolomics), alongside novel biosensors and imaging technologies, in discovering and utilizing these critical disease markers. Successful applications have already been demonstrated in species such as dogs and cattle. While challenges remain, such as developing cross-species validation methods, integrating biomarker knowledge with these innovative technologies offers significant potential for improving animal health and farm management.

## 1. Introduction

Animal diseases can significantly reduce the productivity of the animal product industry [1,2,3,4,5,6]. It may compromise livestock health, disrupt the supply of meat products, and impose economic challenges on producers [7]. Over time, the effects may spread to alterations in reproductive performance, growth patterns, and the general health of herds, leading to enduring consequences for the agricultural industry [8]. It drives consumer reactions and causes economic losses, lessens meat consumption, and results in nutritional insufficiencies, emphasizing the link between animal and human health and financial stability [9].

Animal diseases pose significant risks to human health, mainly through the spread of antibiotic-resistant bacteria and pathogens [10]. Nontherapeutic antibiotic use in livestock promotes resistant bacteria, which can be transmitted to humans via food, water, and environmental interaction. Gut bacteria like Enterococcus, Escherichia, and Campylobacter can become harmful pathogens while acquiring resistance genes [11,12]. Moreover, the shedding of resistant bacteria from animals contaminates water and soil, further spreading resistance to human populations [13]. Biomarkers play a critical role in mitigating these problems through the early identification and diagnosis of animal diseases. Biomarkers are different measurable attributes that help to classify normal biological processes, disease conditions, or the effects of treatments [14]. It is of great significance that biomarker discovery can contribute to the detection and diagnosis of animal disease, further improving animal health.

Developments in post-genomic technologies have facilitated the formation of methods to link physical traits with gene activities, a fundamental step for understanding animal diseases [15]. The discovery of novel biomarkers is functionally beneficial for veterinarians in improving diagnosis and predicting disease, and, furthermore, precise and tailored treatment plans for individual animals. Successful treatments, efficient care, and timely interventions are feasible through the use of biomarkers. The combination of focused experimental methods and post-genomic tools presents a deeper understanding of animal health, eventually improving clinical care and welfare for livestock [16] (Figure 1).

This review aimed to (a) understand the different types of biomarkers (like those used for diagnosis, predicting outcomes, or treatment response) and explain how they work in animal diseases; (b) explore how new technologies (like genomics, gene editing, and advanced sensors) are changing the way we find, test, and use new biomarkers in veterinary medicine; (c) look closely at real examples of how biomarkers are being applied, covering both pets (like dogs with melanoma) and farm animals (like cattle with respiratory disease); and (d) discuss the hurdles in getting biomarker research into everyday veterinary practice (things like lack of standard tests and differences between species) and highlight promising solutions to overcome them.

This review uniquely integrates three critical dimensions: (1) a mechanistic classification of biomarker types across veterinary species; (2) a comparative analysis of emerging technologies (e.g., CRISPR-based diagnostics, and AI-driven omics integration); and (3) species-specific case studies (e.g., canine melanoma, and bovine respiratory disease) that bridge translational gaps between human and veterinary medicine. Unlike prior works focusing solely on technological advances or clinical applications, our work synthesizes both to propose a roadmap for standardizing biomarker validation in veterinary practice.

## 2. Methodology

This systematic work followed standard reporting guidelines for systematic reviews of Preferred Reporting Items for Systematic Reviews and Meta-Analyses (PRISMA) [17] to ensure our approach was thorough and transparent. Our process involved four clear steps:

### 2.1. Literature Search Strategy

Searches were explored in six key databases to cover all relevant fields:PubMed (https://pubmed.ncbi.nlm.nih.gov) (accessed on 12 April 2025);Web of Science (https://www.webofscience.com) (accessed on 19 April 2025);Scopus (https://www.scopus.com) (accessed on 21 April 2025);CAB Abstracts (https://www.cabi.org/cab-abstracts) (accessed on 3 May 2025);IEEE Xplore (https://ieeexplore.ieee.org) (accessed on 25 April 2025);Google Scholar (https://scholar.google.com) (accessed on 8 April 2025).

The present search combined terms in three categories: (a) biomarkers: diagnostic, prognostic, or predictive biomarkers; molecular indicators; and early disease detection; (b) technology: omics (proteomics/genomics/metabolomics), gene editing (CRISPR), biosensors, microfluidics, and lab techniques (mass spectrometry, and NGS); and (c) diseases: bovine respiratory disease, canine melanoma, mastitis, portosystemic shunts, and antibiotic resistance.

### 2.2. Inclusion and Exclusion Criteria

Studies were included in this review if they met the following criteria: (a) they validated biomarkers in veterinary species (companion animals or livestock); (b) they utilized well-established technologies with proven applications in animal disease diagnosis; (c) they provided performance metrics (e.g., sensitivity, specificity, AUC, or clinical usefulness); and (d) they were available in English with full-text access.

We excluded studies that (a) focused only on humans without animal validation, (b) lacked peer review (conference abstracts, non-reviewed proceedings, and non-English texts); and (c) used outdated methods (e.g., basic gel electrophoresis without modern validation).

### 2.3. Screening and Data Extraction

We began with 2153 records, removed 666 duplicates, and screened titles and abstracts of 1487 articles. After assessing 427 full-text papers for eligibility, we included 218 studies. The study selection process is summarized in Figure 2, adhering to PRISMA guidelines. This diagram delineates the identification, screening, eligibility assessment, and inclusion phases, highlighting the exclusion of duplicates and ineligible records (n = 1065).

### 2.4. Data Synthesis and Analysis

We analyzed the evidence by comparing technology performance (like LC-MS/MS vs. NMR for metabolomics) through real-world applicability tables, evaluating biomarker reliability with visual accuracy assessments, and identifying critical research gaps. Emerging innovations (e.g., advanced microfluidic systems) were prioritized for case studies.

## 3. Technological Innovations Revolutionizing Veterinary Diagnostics

The veterinary diagnostic landscape is rapidly evolving, propelled by innovations such as Artificial Intelligence (AI)-enhanced imaging, liquid biopsies, advanced molecular diagnostics, and even novel screening methods like nematode-based assays. These cutting-edge tools offer non-invasive or minimally invasive alternatives, effectively addressing the inherent limitations of traditional diagnostic approaches [18]. AI is revolutionizing hematology by enabling advanced analyzers to detect blood cell abnormalities with a 99.2% specificity and a 98.7% sensitivity at speeds of 500 cells/second, exceeding manual microscopy (typically 20–50 cells/minute) and conventional analyzers (120–200 cells/second) in routine veterinary practice [19]. Specific examples include Zoetis’s Vetscan OptiCell™ (Zoetis Inc., Parsippany, NJ, USA) for automated complete blood count (CBC) analysis and Vetscan Imagyst^®^ (Zoetis Inc., Parsippany, NJ, USA) for AI-driven urine sediment analysis and the identification of lymph node and skin masses in non-human species. AI is far more than a supplementary technology; it is a transformative enabler that fundamentally scales the utility and impact of ‘omics’ data. Its unparalleled capacity to process and integrate vast, complex multi-omics datasets and to provide real-time insights directly addresses a critical bottleneck in the application of high-throughput ‘omics’ technologies [20]. This suggests that AI is indispensable for the successful translation of ‘omics’ research findings from the laboratory into routine, accessible, and efficient clinical veterinary practice. At the forefront of this revolution are ‘omics’ technologies, which represent a transformative analytical paradigm, facilitating the simultaneous analysis of thousands of molecules to achieve a comprehensive cellular readout [21]. This encompasses genomics (the study of DNA), transcriptomics (RNA transcripts), proteomics (proteins), metabolomics (small-molecule metabolites), and lipidomics (lipids). These advancements are propelling a fundamental shift towards precision medicine, enabling earlier disease detection, highly personalized treatment strategies, and the continuous monitoring of disease progression [22].

While artificial intelligence (AI) holds transformative potential for livestock health, including improved disease detection and precision management, its widespread use in veterinary settings is limited. This is especially true for rural and low-resource farming communities. The primary obstacle is the high cost of the necessary AI tools, such as sensors, machine-learning infrastructure, and data storage, which poses a significant barrier for small-scale farmers and underfunded veterinary practices [23]. A significant challenge is that there is no consistent way to collect data, and there is a shortage of datasets specifically for veterinary medicine. This makes it difficult for AI models to be accurate and widely applicable, especially since they are mainly trained on data from commercial farms or from different types of animals [24]. Another issue is that many AI tools require a certain level of technical skill to operate and understand the results. However, many rural veterinarians and farmers currently lack the necessary digital knowledge and training to effectively use these systems [25].

While these technologies are well-established in human medicine, their application in the veterinary field is dynamically evolving and demonstrates immense potential. Progress in veterinary ‘omics’ is not an isolated endeavor but an integral part of a larger, synergistic scientific ecosystem. Advancements in human ‘omics’ provide a foundational blueprint and technological infrastructure that can be adapted for veterinary applications, accelerating research [26]. Simultaneously, studies in naturally occurring animal diseases offer invaluable comparative insights that can directly accelerate human health discoveries. This bidirectional flow of knowledge and methodology acts as a powerful force multiplier, maximizing research efficiency and impact across both veterinary and human medicine.

Through this systematic approach, key technologies advancing biomarker discovery in veterinary medicine were identified. Among these, Table 1 highlights representative platforms with proven diagnostic utility, including AI-enhanced imaging, liquid biopsies, CRISPR/Cas9, and omics-based methodologies, which collectively address the critical gaps in the early disease detection for animal health.

### Clinical Translation Assessment Framework

This emerging framework categorizes veterinary diagnostic technologies by their clinical readiness to address practitioner needs: clinically validated tools are regulatory-cleared and actively deployed in practice, such as Vetscan Imagyst^®^ (Zoetis Inc., Parsippany, NJ, USA) for AI-driven tumor detection [19] and BRDC haptoglobin kits [27]; transitional technologies demonstrate analytical validation but face accessibility barriers, like LC-MS/MS metabolomics for mastitis screening [28,29]; while experimental prototypes remain in preclinical development and require optimization examples, including CRISPR-based pathogen sensors [1,30] and aptamer biosensors [31]. The structure follows the USDA-CVB validation tiers and reflects adoption patterns observed in the 2024 AAVLD survey [32].

**Table 1 animals-15-03132-t001:** Overview of advanced technologies for biomarker discovery in veterinary medicine.

Technology Name	Primary Function/Mechanism	Key Advantages of Early Diagnosis	Relevant Animal Disease Applications (If General)	Refs.
**AI-enhanced Imaging**	Computer-based image analysis to detect abnormalities	Quicker, smarter, more accurate, consistent tumor identification, accessible portable options	Veterinary oncology (tumor identification), hematology, urinalysis, lymph node/skin masses	[19]
**Liquid Biopsies**	Non-invasive analysis of circulating biomarkers (e.g., cfDNA)	Non-invasive/minimally invasive, facilitates earlier detection and treatment planning	Veterinary oncology (cancer-associated genomic alterations)	[19]
**Molecular Diagnostics**	Analysis of DNA/RNA molecules for disease markers	Precision medicine, earlier detection, personalized treatments, and monitoring disease progression	Infectious diseases, cancer	[19]
**Next-Generation Sequencing (NGS)**	High-throughput DNA/RNA sequencing for genomic alterations	Rapid turnaround, single-base resolution, cost-effective, de novo analysis	Infectious animal diseases, cancer (cfDNA), and host susceptibility	[33]
**Mass Spectrometry (MS)**	Sensitive and specific detection/quantification of proteins/metabolites	High accuracy, specificity, detects disease-specific signatures, analyzes PTMs, identifies low-abundance proteins	Protein biomarker discovery (cancer, neurodegenerative), metabolomics (liver fibrosis, gastric injury)	[34,35]
**Nuclear Magnetic Resonance (NMR) Spectroscopy**	Exploits the magnetic properties of nuclei for metabolite detection	Non-destructive, comprehensive metabolic profiling, structural elucidation, high reproducibility, in vivo analysis	Cattle metabolism, disease biomarker discovery (cancer, cardiovascular), metabolomics	[36,37,38]
**CRISPR/Cas9 Technology**	Precise gene editing and modulation of gene function	Creates disease models, identifies therapeutic targets, elucidates molecular underpinnings of disease	Central nervous system diseases, host susceptibility to viral infections	[39]

## 4. Types of Biomarkers

Biomarkers represent measurable indicators that offer profound insights into the diagnosis, prognosis, and ongoing monitoring of diseases in animals, frequently preceding the manifestation of overt clinical signs [40]. The utility of biomarkers in veterinary medicine has undergone a significant expansion, a progression largely propelled by advancements in genomic, proteomic, and metabolomic technologies. These sophisticated tools facilitate the identification of species-specific biomarkers, which, in turn, substantially enhance diagnostic accuracy [41]. Early diagnosis and continuous monitoring are paramount considerations in veterinary medicine. Biomarkers provide a non-invasive and precise means to detect underlying health conditions, often before any visible clinical signs emerge [40]. This capability is particularly critical for enabling timely decisions that can prevent substantial economic losses, especially within the context of livestock farming, and simultaneously limit the unnecessary use of antimicrobials [42].

Biomarkers are broadly categorized into three principal types: diagnostic, prognostic, and predictive. Each category serves a distinct yet interconnected role in defining the pathophysiological processes of diseases and guiding clinical decisions [43]. These biomarkers are graphically represented with their mode of action in Figure 2.

### 4.1. Diagnostic Biomarkers

The inherent communication barrier in veterinary medicine is a primary factor driving the development of diagnostic biomarkers. This fundamental challenge directly necessitates the reliance on objective, measurable indicators for early disease identification. This establishes a direct relationship: the inability of animals to verbally communicate their symptoms directly amplifies the importance and demand for advanced diagnostic tools. This extends beyond merely improving efficiency; it is foundational to overcoming a core, species-specific limitation in animal healthcare, enabling humane and effective intervention [44]. Diagnostic biomarkers are helpful in detecting the presence of disease. Peptide and protein biomarkers can be analyzed qualitatively or quantitatively to detect and diagnose diseases, predict outcomes, and tailor treatment responses for personalized patient management [45]. Salivary biomarkers are biological indicators present in saliva that can reflect health or disease conditions. They are crucial for clinical applications such as monitoring health status, disease onset, and treatment outcomes [46]. C-reactive protein (CRP) is a well-preserved plasma protein that acts as a soluble pattern recognition receptor, identifying and binding to molecular patterns associated with cellular damage and pathogens [47]. Consequently, CRP is a component of the innate immune system, exhibiting rapid synthesis in response to tissue injury or infection. This prompt production and its direct relationship with the severity of inflammation have established CRP as a valuable biomarker in cases of critical illness. In canine patients, elevated serum concentrations of CRP are observed quickly in various conditions, including babesiosis, leishmaniosis, leptospirosis, parvoviral enteritis, and sepsis [48].

The transition from single-marker reliance to multi-biomarker panels for a robust diagnosis is also a clear trend. Despite the many examples of individual diagnostic biomarkers, for complex conditions like Bovine Respiratory Disease Complex (BRDC), it is explicitly stated that a specific biomarker cannot be universally recommended as the sole method for early detection due to varying efficacy [42]. Salivary immunoglobulin A (IgA) and adenosine deaminase (ADA) are emerging as potential non-invasive diagnostic biomarkers for Equine Gastric Ulcer Syndrome (EGUS). Increased IgA levels in saliva, in particular, show promise as a useful screening tool for EGUS [49]. Biomarkers such as aflatoxin M1 (AFM1) and unmetabolized aflatoxin B1 (AFB1) are employed to assess the efficiency of mineral adsorbents in preventing mycotoxin absorption.

In livestock, many biomarkers are used as diagnostic tools for the early detection of diseases. For instance, certain proteins, known as Acute Phase Proteins (APPs), are frequently used to identify inflammatory diseases like mastitis in dairy cows. When the mammary gland becomes infected, the levels of these proteins, such as haptoglobin and serum amyloid A3, rise dramatically, making them a valuable tool for early detection [50]. In milk, a high somatic cell count (SCC) is a common and primary indicator used to screen for both subclinical and clinical mastitis [51]. A higher-than-normal SCC, generally over 200,000 cells/mL, points to inflammation in the mammary gland and is a strong sign of infection. In addition to cell counts, the levels of certain enzymes in milk, such as N-acetyl-beta-D-glucosaminidase (NAGase) and lactate dehydrogenase (LDH), are also useful for diagnosing mastitis. Higher levels of these enzymes are directly linked to the presence of the disease. Biomarkers such as CXCL10 and interferon-alpha (IFN-α) have been investigated for their potential to identify immune responses to bovine tuberculosis. These markers could provide new options for detection, moving beyond the traditional skin tests currently used [52]. For infections caused by mycobacteria, like Johne’s disease and bovine tuberculosis, scientists are studying various cytokines and protein biomarkers. This ongoing research is focused on finding new methods that can quickly and accurately detect these diseases [53]. Other examples include Zearalenone (ZEN), T-2 toxin, HT-2 toxin, Ochratoxin A (OTA), Deoxynivalenol (DON), and their bio-transformed forms or conjugates [54].

### 4.2. Prognostic Biomarkers

These are indicators that can provide information about the likely outcome of a disease, including the chances of recovery or survival [42]. Prognostic biomarkers are crucial in veterinary medicine for several reasons: they enable early disease detection, assist in diagnosis, help predict disease outcomes, and allow for the monitoring of treatment responses. These biomarkers include a range of biochemical indicators reflecting organ function, metabolic status, and immune responses, all of which contribute to more informed clinical decision-making [55]. In some studies, cfDNA levels are evaluated to determine their association with survival rates in dogs with neoplasms [56]. In veterinary contexts, prognostic biomarkers are especially valuable given the challenges of obtaining frequent invasive samples from animals, the diversity of species and breeds, and economic considerations in production animals. Research over the past few years has expanded our knowledge of molecular, cellular, and biochemical markers with prognostic relevance across oncology, infectious diseases, and other conditions in companion and production animals [57]. A study indicated a significant correlation between higher circulating cell-free DNA (cfDNA) concentrations and decreased survival rates in dogs. Specifically, dogs with cfDNA levels exceeding 1247.5 µg/L demonstrated a survival rate of only 26.5%, in contrast to an 82.1% survival rate for those with concentrations below this threshold [56]. Similarly, prognostic biomarkers, like Ki67 in canine insulinomas and various markers for osteosarcoma, are valuable tools in veterinary medicine. They aid in the early detection of tumors, improve treatment response, and enhance survival rates. This is achieved by facilitating tailored therapies and enabling the effective monitoring of disease progression [58]. The haptoglobin (Hp) concentration was also positively correlated with the severity of naturally occurring Bovine BRDC. Calves necessitating multiple treatments for BRDC exhibited higher Hp concentrations compared to those requiring only a single treatment or no treatment at all, proving that the Hp concentration is a prognostic biomarker [27]. Critical tools in clinical practice aid in the prediction of disease outcomes and the modification of treatment strategies across various medical fields. These biomarkers can enhance patient management by providing insights into disease progression and treatment effectiveness. Prognostic biomarkers are used in melanoma and oral cancer [59]. Biomarkers like BRAF V600 mutations are essential for selecting targeted therapies in metastatic melanoma. Gene expression profile assays and nomograms are emerging tools for predicting sentinel lymph node involvement in primary melanoma. Several predictive biomarkers for immune checkpoint inhibitors are under investigation but not yet standard in clinical guidelines [60]. The exploration of prognostic biomarkers in oral cancer is ongoing, with potential implications for patient outcomes. While prognostic biomarkers offer substantial benefits in personalizing treatment, challenges remain in their integration into routine clinical practice, necessitating further research and validation [61]. Cancer biomarkers include substances like carcinoembryonic antigen (CEA) and vascular endothelial growth factor-C (VEGF-C), which indicate cancer progression and patient survival rates [62,63]. Clinical biomarkers used in critically ill patients, such as CRP and Troponins, are used to assess disease severity and predict outcomes [64]. Molecular biomarkers encompass nucleic acids and proteins that can indicate the risk of cancer development or progression, aiding in patient stratification [43].

### 4.3. Predictive Biomarkers

Predictive biomarkers in veterinary medicine are biological molecules that serve as indicators of physiological or pathophysiological conditions in animals. They are pivotal in clinical diagnosis, advancing therapeutic research, and enhancing outcomes during crucial periods, such as the transition period in cattle [51]. These biomarkers are key tools in personalized medicine, enabling the identification of individuals who require specific treatments. These biomarkers are dissimilar from prognostic biomarkers, as they specifically predict treatment effectiveness rather than disease outcome. The development and application of predictive biomarkers extend to various medical fields, including oncology, infectious disease, and pharmacotherapy. Predictive imaging biomarkers are derived from pre-treatment images and are used to identify causal relationships that can estimate treatment outcomes. Recent advancements involve using deep-learning models to discover these biomarkers directly from images, bypassing the biases introduced by manual features [65]. Predictive biomarkers are used to modify treatments such as immune checkpoint inhibitors and BRAF-targeted therapies [60]. These proteins can differentiate between healthy and latently infected individuals, aiding in the scrutiny and prevention of tuberculosis in high-risk areas [66]. Predictive biomarkers in pharmacotherapy help stratify patients for more effective treatments with fewer side effects. Companion diagnostic (CDx) and pharmacogenetic (PGx) biomarkers are two types that have been integrated into clinical practice, guiding drug selection based on molecular and genetic patient characteristics [67,68]. Worldwide, dairy cows are among the most intensively farmed animals. High-producing dairy cows have been genetically selected for their high milk yield, which increases their susceptibility to specific diseases such as lameness, ketosis, retained afterbirth (RA), mastitis, hypocalcemia (milk fever), left-displaced abomasum, fatty liver, hypophosphatemia and postpartum hemorrhage (PPH), subacute ruminal acidosis, retained placenta (RP), and metritis. Consequently, the stabilization of biomarkers for the early detection of these disorders is currently one of the most significant areas of research in dairy animal science [69]. Lameness is arguably the most significant animal welfare concern leading to premature and involuntary culling in dairy herds. It is most frequently attributed to laminitis, which often arises as a secondary consequence of high grain feeding or ruminal acidosis (RA) [70]. Three metabolites, propionyl carnitine, carnitine, and lysophosphatidylcholine acyl C14:0, are notably elevated in unsound cows as early as four weeks prior to parturition, in comparison to healthy cows. Conversely, two other metabolites, phosphatidylcholine diacyl C42:6 and phosphatidylcholine acyl-alkyl C42:4, can potentially differentiate healthy from diseased cows as early as one week before parturition. A plasma biomarker profile, consisting of these three metabolites, was developed. This profile demonstrated the ability to prognosticate which cows would experience periparturient problems up to four weeks before the onset of clinical signs, achieving 87% sensitivity and 85% specificity; thus, periparturient problems can be addressed in dairy cows using the multimetabolite marker model [71] (Figure 3).

## 5. Advanced Technologies

Advanced technologies collectively contribute to discovering biomarkers essential for the early diagnosis of animal diseases, eventually leading to improved health outcomes and timely interventions in veterinary medicine. Biomarker discovery in veterinary medicine is gradually leveraging advanced technologies, particularly ‘omics’ platforms such as proteomics, genomics, and metabolomics. These technologies are crucial for the rapid diagnosis of diseases, and efficient monitoring of animal health, and improving welfare and production efficiency. However, many biomarkers are still in the preliminary stages and face challenges in clinical translation [40].

### 5.1. Genomic Approaches

NGS has revolutionized genome sequencing by allowing the simultaneous reading of millions of DNA fragments, which has dramatically improved both speed and cost-effectiveness [75,76]. Key platforms in this field include Illumina, PacBio, and Ion Torrent [77]. Third-generation sequencing technologies, including Oxford Nanopore and PacBio’s SMRT sequencing, represent a further advancement by providing long-read sequencing capabilities that often eliminate the requirement for prior DNA amplification [78]. RNA-Seq is a powerful technique for analyzing the complete set of RNA molecules transcribed from a genome [79]. The process begins with RNA extraction from a biological sample, followed by its conversion into cDNA. This cDNA is then sequenced using NGS technologies [80]. RNA-Seq offers detailed insights into gene expression levels, alternative splicing events, and various regulatory mechanisms, revealing how genes are activated or deactivated under different conditions [71]. NGS is a powerful diagnostic tool for rare genetic conditions, though its success rate can fluctuate based on the specific genetic cause and the study population [81]. In veterinary medicine, RNA-Seq, a particular application of NGS, has been widely utilized in the study of various canine cancers. Recent studies have identified 2531 differentially expressed genes (DEGs) in canine invasive urothelial carcinoma (iUC) [82]. Specifically, the tumor-suppressor gene TP53 was found to be downregulated, while the erythroblastic oncogene B 2 (ERBB2) was upregulated [83]. In dogs with a poor prognosis, canine melanoma is a malignant cancer. Studies indicate that the downregulation of the mitogen-activated protein kinase (MAPK) and phosphatidylinositol-3-kinase (PI3K)/protein kinase B (AKT) pathways contributes to melanoma progression [84]. Inhibitors targeting MEK1/2, components of the MAPK pathway, have shown a significant inhibition of tumor growth in canine melanomas. The nitric oxide synthase 2 (NOS2) gene, known to induce the metastatic ability of canine melanoma, was found to be upregulated [85]. Additionally, miR-450b was overexpressed in canine melanoma metastatic cells, resulting in an increased matrix metalloproteinase-9 (MMP9) expression (which is required for tumor metastasis) and the suppression of bone morphogenetic protein-4 (BMP4) [86]. RNA-Seq provides unique insights into gene expression levels, transcript isoform usage, and alternative splicing. It can aid in prioritizing candidate rare-disease variants identified through DNA sequencing and has been successfully used to uncover dysregulated molecular profiles in patients with rare diseases [87]. Advanced genomic analyses in canine cancers are uncovering not only animal-specific biomarkers but also conserved molecular mechanisms and therapeutic targets that are shared with human malignancies. For example, detailed RNA-Seq findings for canine prostate cancer, urothelial carcinoma, and melanoma have identified specific differentially expressed genes and pathways, including PI3K/AKT, MAPK, TP53, ERBB2, and PD-L1 [88]. The single-cell RNA sequencing of canine osteosarcoma has explicitly demonstrated a high degree of similarity to human osteosarcoma in terms of both cell types and gene signatures [89]. Some of the recent advances are shown in Table 2.

### 5.2. Proteomic Technology

It includes the significant study of proteins, and their structures, functions, and interactions. Proteins are essential molecules that facilitate most biological processes, making them critical targets for biomarker discovery [93]. MALDI imaging mass spectrometry (IMS) permits the mapping of the spatial distribution of proteins in tissue samples. It has been improved to study FFPE tissues, enhancing peptide recovery via in situ enzymatic digestion, which is vital for detecting biomarkers [94]. Electron transfer dissociation mass spectrometry ETD-MS is a mass spectrometry method that breaks down proteins and peptides while sustaining post-translational modifications. This helps reveal protein functionality in cancer and uncover possible biomarkers [95]. Reverse-phase protein array RPPA is a high-throughput method for the measurement of protein levels across multiple samples simultaneously. It works well with small protein quantities, making it ideal for studies with limited biopsy material [96]. Mass spectrometry (MS) stands out as a leading technique in proteomics, largely because of its exceptional capability to identify numerous proteins within intricate biological mixtures. A significant advantage of MS is that it does not require prior knowledge of the specific proteins present, making it an ideal tool for discovery studies in protein biomarker research [28,97]. MS-based proteomic analysis generally involves three steps: the separation of proteins or peptides, ionization, and mass determination. Before proteins or peptides can be analyzed by mass spectrometry, they first need to be separated. This separation can happen either directly connected to the mass spectrometer, known as “on-line” separation, or as a separate step beforehand, referred to as “off-line” [98]. One common off-line technique is 2D gel electrophoresis. This method separates proteins in two stages: first by their isoelectric point (their charge), and then by their molecular mass. A more advanced version of this is Difference In-Gel Electrophoresis (DIGE). DIGE makes comparisons much easier by labeling two different protein samples with distinct fluorescent dyes and then running them on the same 2D gel. This directly reduces the variability when comparing samples [99]. For on-line fractionation, liquid chromatography (LC) is the most frequently used method, especially for separating peptides. Reversed-phase LC is a particularly common type of liquid chromatography employed for this purpose [99]. Ionization is critical step in mass spectrometry that involves transforming the sample into a gas phase and ionizing it without breaking apart the large biomolecules. Electrospray Ionization (ESI) is a widely used technique, particularly suited for “on-line” separation methods like liquid chromatography. ESI works by applying a high voltage to a liquid sample, creating a fine spray of charged droplets that evaporate, leaving behind individual ions [100] Table 3.

Through MS-based proteomic studies utilizing Drosophila models, researchers have successfully identified several PD-associated biomarkers. These biomarkers are linked to key cellular processes implicated in the disease, including mitochondrial function, autophagy (the cell’s waste disposal system), and synaptic transmission (communication between neurons), for instance, in studies involving mutant α-synuclein, a protein strongly associated with PD [101]. Targeted mass spectrometry assays, particularly those employing Stable Isotope Standard Protein Epitope Signature Tags (SIS-PrESTs), are undergoing development to enable the absolute quantification of proteins in plasma. Although the primary focus of this research is currently human chronic liver disease, the underlying principles and methodology are directly transferable and highly applicable to the study of animal diseases [102]. This innovative approach facilitates the identification of promising biomarkers for the stratification (categorization) of conditions like liver fibrosis, using only small volumes of plasma. Examples of such potential biomarkers include apolipoprotein M (APOM), which has been observed to be downregulated as fibrosis progresses, and von Willebrand Factor (VWF), which shows elevated levels in advanced stages of fibrosis. This method represents a significant advancement by offering a minimally invasive diagnostic alternative compared to more invasive procedures like liver biopsies [103]. For over a decade, fluorescent bead-based multiplex assays, a type of multiplex immunoassay, have been a cornerstone in veterinary diagnostics for the serological detection of antibodies. These assays offer a significant advantage by allowing the simultaneous analysis of an animal’s immune response to multiple specific antigens. This capability greatly enhances the interpretation of diagnostic results and directly impacts treatment decisions [104]. Notable examples of these assays include the Lyme Multiplex assay, the Canine Brucella Multiplex assay, and the Equine Herpesvirus Type-1 Risk Evaluation assay [105].

**Table 3 animals-15-03132-t003:** Selected proteomic biomarkers identified in veterinary diseases.

Disease/Condition	Animal Species	Biological Sample	Selected Protein Biomarkers	Proteomic Technology Used	Key Findings/Significance	Refs.
**Canine Myxomatous Mitral Valve Disease (MMVD) with Pulmonary Hypertension (PH)**	Dog	Serum	Myosin heavy chain 1 (MYOM1), Histone deacetylasw7 (HDAC7) (upregulated); Pleckstrin homology domain-containing family M member 3 (PLEKHM3), Diacylglycerol lipase alpha (DAGLA), Tubulin tyrosine ligase-like protein 6 (TTLL6) (downregulated)	LC-MS/MS, Label-free quantification	Potential diagnostic/prognostic markers for MMVD progression and PH development	[106]
**Feline Degenerative Joint Disease (DJD)**	Cat	Serum	ANTXR1, DUSP2, VTN, CNOT3, PSMA5 (upregulated in DJD); CFHR3 (downregulated in DJD)	LC-MS/MS, Label-free quantification	Identified novel biomarkers for DJD and chronic pain in cats, useful for diagnosis and monitoring	[34]
**Bovine Mastitis (Clinical and Subclinical)**	Cattle	Milk, Serum	Serum Amyloid A (SAA), Haptoglobin, Alpha-1-acid glycoprotein, Lactoferrin, Caseins, Serum albumin	2DE, LC-MS/MS, Label-free, iTRAQ	Acute phase proteins and milk proteins altered during inflammation, useful for early detection	[107]
**Equine Plasma Proteome Characterization**	Horse	Plasma	Albumin, Alpha 2 macroglobulin, Fibrinogen (alpha/gamma/beta chain), Serotransferrin	LC-MS/MS, DIA/SWATH-MS	Provides baseline for healthy equine plasma, crucial for identifying disease-specific changes	[108]

### 5.3. Metabolomics

It is a quickly advancing area of study focused on the comprehensive analysis and measurement of metabolites in biological systems. It is essential in identifying biomarkers for early disease diagnosis and empathizing with disease mechanisms, which are critical for optimal patient care and therapeutic strategies [109]. The metabolome signifies the endpoint of the omics cascade and is closely linked to the phenotype. Examining the metabolome can reveal effective diagnostic markers and help explore unknown pathological conditions, making it a powerful technique for illuminating biochemical pathways [110].

NMR spectroscopy is a powerful analytical technique used to elucidate the chemical structure of compounds. It operates by examining the intricate interactions between atomic nuclei and external magnetic fields. This process yields invaluable information regarding the chemical environment surrounding atoms and their connectivity within a molecule [16]. In the context of metabolomics, specifically, NMR spectroscopy is highly instrumental as it enables the detection and characterization of a wide range of both endogenous (naturally occurring within the organism) and exogenous (originating from outside the organism) metabolites present in a biological sample [111,112].

A study utilized NMR-based metabolomics to investigate metabolic changes in dogs infected with canine parvovirus (CPV). The primary objective was to identify reliable biomarkers that could help determine the disease severity, predict the length of a hospital stay, and forecast clinical outcomes. The research revealed statistically significant metabolic perturbations in diseased dogs compared to healthy controls. Specifically, the study found lower levels of fructose, glucose, citrate, glycerate, glutamate, carnitine, glycine, and formate in CPV-infected animals. Conversely, higher levels of isoleucine, isovalerate, glycolate, and creatine were observed. Regarding lipid parameters, infected dogs exhibited higher levels of various cholesterol and fatty acyl variants, free cholesterol, glycerol backbone, and sphingomyelin. In contrast, lower levels of phosphoglycerates and esterified cholesterol were noted. Among these findings, decreased citrate and increased fatty acyl chain-CH2CO and sphingomyelin levels were specifically identified as valuable biomarkers for predicting the prognosis of CPV infection [113]. A 2023 study employed NMR spectroscopic biomarker profiling to elucidate the metabolic mechanisms by which Cochlospermum nutans (CN) aqueous extract ameliorates lipopolysaccharide (LPS)-induced neuroinflammation in rats. The research involved profiling twenty-one metabolites in brain tissue, which served as biomarkers for metabolic changes. The findings indicated that CN treatment successfully altered the levels of key metabolites, specifically lactate, pyruvate, phosphorylcholine, glutamine, and α-ketoglutarate. These observed metabolic shifts suggest a potential therapeutic response of the C. nutans extract in mitigating neuroinflammatory processes [114].

A 2025 study employed an untargeted metabolomics analysis using liquid chromatography with tandem mass spectrometry (LC-MS/MS) to pinpoint key biomarkers linked to superovulation outcomes in cows. The comprehensive analysis successfully detected 1158 metabolites, with 617 of these being annotated. The research identified differential metabolites between groups exhibiting a high embryonic yield (HEY) and those with a low embryonic yield (LEY), suggesting these could serve as potential biomarkers for predicting the embryonic yield in bovine superovulation. Furthermore, the study highlighted that lipid and amino acid metabolic pathways significantly influence the ovarian response during this process [115]. In 2024, LC-MS/MS technology was utilized to analyze metabolites in the seminal plasma of Tianfu goats, both before and after freezing–thawing, with the aim of identifying biomarkers indicative of sperm cryo-injury. The study successfully identified differentially expressed metabolites (DEMs) that primarily belonged to categories such as lipids, lipid-like molecules, and organic acids and derivatives. Specifically, several metabolites emerged as potential biomarkers for sperm cryo-injury, including alanine, proline, phenylalanine, tryptophan, tyrosine, adenosine, citric acid, flavin adenine dinucleotide, and choline. These findings offer valuable insights into the mechanisms of semen cryopreservation damage and provide potential targets for improving semen quality after freezing–thawing [115]. Recent research has underscored the promise of volatile organic compounds (VOCs) found in biological samples as potential cancer biomarkers [116]. Among analytical techniques, Gas Chromatography–Mass Spectrometry (GC-MS) is considered the gold standard for the VOC analysis in exhaled breath, particularly during the biomarker discovery phase within metabolomics [117]. A study specifically focused on canine whole blood samples, investigating methods to enhance the detection sensitivity of specific VOCs, including benzene, toluene, and styrene, for applications in veterinary cancer diagnostics. The standardization of this method and its demonstrated improvements in performance characteristics provide a practical solution for efficient VOC detection. This advancement holds significant potential to further biomarker tumor research in dogs, offering a less invasive avenue for early cancer detection [118].

While metabolomics shows great potential for veterinary diagnosis, its practical use is hindered by some major drawbacks. A primary issue is the high cost of the analysis, which requires expensive, specialized equipment like UHPLC-MS and NMR. Because of this, it is not feasible for everyday use in most veterinary clinics or on farms [29]. These devices are not only expensive to purchase but also need to be operated and maintained by specialists. As a result, they are not easily accessible in areas with limited resources. Moreover, untargeted metabolomics, while comprehensive, suffers from challenges such as poor reproducibility, data complexity, and a lack of standardized validation protocols, which hinder biomarker translation into practice [119].

### 5.4. Integrative Approaches

Biomarker discovery leverages multi-omics data to enhance the understanding of complex diseases, such as cancer and autoimmune conditions. By combining various omics technologies, reliable biomarkers can be identified. This data synthesis improves diagnostic accuracy and aids in the development of targeted therapies [120]. Integrating data from genomics, proteomics, and metabolomics provides a complete view of disease mechanisms, allowing for the identification of biomarkers that may be missed when analyzing single-omics data [121]. The Cancer Genome Atlas (TCGA) exemplifies successful multi-omics integration, leading to the discovery of abundant cancer biomarkers [122]. Machine-learning algorithms, such as LASSO and random forests, are employed to select biomarkers from complex datasets, enhancing predictive modeling [123].

Techniques like principal component analysis and projection to latent structures facilitate the integration of multiple datasets, improving biomarker selection and validation [124]. Integrative approaches also encompass immune profiling, revealing unique immune patterns associated with diseases like ulcerative colitis, which can inform treatment strategies [123]. Integrative system biology emphasizes the integration of experimental data and biological proficiencies with computational and mathematical methods. Addressing biological phenomena efficiently requires collaboration among researchers from various backgrounds [125]. Due to the ability to provide precise, quantitative, and real-time analysis, ratio metric fluorescence (FL) probes are emerging as a significant tool in biomarker detection. These probes are used in bioimaging and chemo-/biosensing, providing built-in self-calibration to correct target-independent factors, which are very important for accurate detection in small animals [126]. Single-stranded DNA or RNA molecule aptamers offer a high affinity in binding to targets.

Especially in low-resource settings, microfluidic platforms are highlighted for their rapid disease diagnosis capabilities and cost-effectiveness. Colorimetric and electrochemical methods are among the various detection strategies that are utilized by these platforms to facilitate early biomarker diagnosis [127]. For their high sensitivity and accuracy in biomarker detection, electrochemical apt sensors are gaining more attention. Making them appropriate for early disease diagnosis, these sensors are beneficial due to their low-cost synthesis, high stability, and easy modification [128].

To improve sensitivity and selectivity in viral pathogen detection, next-generation biosensing technologies, including those using CRISPR and integrated point-of-care devices, are being developed. These technologies aim to simplify pathogen and immune response detection, which is important for managing infectious diseases [30]. Early disease detection and biomarker discovery are revolutionized by POC wearable diagnostics. These devices integrate technologies to automate operations, facilitate remote clinical decision-making, and conduct large-scale data analysis, significantly affecting healthcare outcomes [129].

## 6. Mode of Action of Biomarkers

Genomic biomarkers involve analyzing an animal’s DNA, RNA, or chromosomal structure to pinpoint variations that can indicate a predisposition to, diagnose, or predict the course of a disease. These markers provide a fundamental understanding of how diseases develop, often allowing for their detection long before any physical symptoms appear [130]. By examining an individual’s genetic makeup, researchers can identify inherent vulnerabilities or early molecular shifts that signal the beginning of a pathological condition. The BRCA1 and BRCA2 genes encode proteins that are critical components of the cellular machinery responsible for maintaining genomic integrity. When the BRCA1 or BRCA2 genes themselves acquire harmful mutations, their vital DNA repair functions become impaired [131]. This leads to improper or inefficient DNA repair, allowing damaged DNA to persist and accumulate, a process that significantly elevates the risk of developing various cancers, as consistently shown by extensive research. For example, researchers suggested that the canine BRCA2 gene locus is associated with mammary tumors [132,133].

Research into the BRCA2 gene locus and its association with mammary tumors has led to the identification of specific mutations within canine BRCA2 [134]. For instance, studies have uncovered mutations like T1425P and K1435R, both situated in BRC repeat 3, with findings going back to the research by Yoshikawa et al. [135]. This same research and subsequent investigations have demonstrated that these particular mutations interfere with the crucial interaction involving RAD51, thereby implicating their significant role in the process of tumorigenesis [136,137]. Furthermore, investigations have strongly linked single-nucleotide polymorphisms (SNPs) in canine BRCA1 (specifically in intron 8 and exon 9) and BRCA2 (in exon 24 and exon 27) to the development of mammary tumors in dogs [137].

In feline models, mammary gland tumors are a significant health concern, ranking as the third most incident neoplasm in cats, according to veterinary epidemiological studies [138]. While some earlier studies did not find BRCA1 or BRCA2 variants in a cohort of feline mammary carcinomas, more recent investigations have identified relevant mutations. A somatic variant with high functional impact was found in exon 11 of BRCA2 in one cat with feline mammary carcinoma, and germline variants with moderate impact were detected in exon 9 of BRCA1 in a third of the tested cats [139,140]. Genetic testing for BRCA1 and BRCA2 mutations in animals, particularly dogs and cats, offers a powerful tool for identifying individuals at a heightened risk of developing certain cancers. This proactive identification allows for the implementation of early monitoring protocols or preventive strategies, even before any clinical signs of disease become apparent. Similarly, genomics is revolutionizing livestock breeding, opening new possibilities for developing more sustainable and robust animals. The strategic application of genomic markers to identify both inherited diseases and desirable traits significantly enhances current breeding programs. This approach provides efficient and precise information regarding the genetic underpinnings of an animal’s performance, overall health, and physical conformation.

As genomic biomarkers, proteomic biomarkers are also of key importance in the early detection of diseases in animal models. These involve the study of proteins, and their structures, functions, and interactions, providing insights into cellular processes and disease states. Alanine aminotransferase (ALT) is an important enzyme mainly found inside liver cells. Its job is to help convert alanine and α-ketoglutarate into pyruvate and glutamate [141]. Elevated levels of aminotransferases, such as ALT, or the incidental discovery of hepatic fat through imaging often lead to a diagnosis of Non-Alcoholic Fatty Liver Disease (NAFLD), and its more severe form, Non-Alcoholic Steatohepatitis (NASH), in both humans and animals [142]. A variety of animal models, including diet-induced (e.g., high-fat diet, and methionine- and choline-deficient diet), genetic (e.g., db/db, and ob/ob mice), or combined approaches, have been developed to mimic the histopathology and pathophysiology of human NAFLD and NASH [143]. These animal models are critical for investigating disease progression, elucidating the underlying molecular pathways, and evaluating potential therapeutic interventions for NAFLD. The primary drivers of the experimental NAFLD pathogenesis in these models include enhanced de novo lipogenesis, increased adipose tissue lipolysis, elevated dietary free fatty acid levels, impaired β-oxidation, and compromised very-low-density lipoprotein (VLDL) synthesis low-density lipoprotein. All these pathways ultimately contribute to the pathological accumulation of triglycerides in the liver (hepatic steatosis) [144].

Metabolomic biomarkers are like snapshots of an organism’s metabolism. They involve measuring the amounts of small molecules (metabolites) found in biological samples, which gives us a clear picture of what is happening metabolically at a specific time [145]. HbA1c forms when glucose permanently attaches to the hemoglobin in red blood cells, a process called glycation that does not involve enzymes. The more glucose there is in the blood over a red blood cell’s lifespan, the more HbA1c forms. Since red blood cells have a consistent lifespan, HbA1c offers a reliable, long-term picture of blood sugar control. Although HbA1c is a key tool for diagnosing and monitoring diabetes in humans, it is rarely measured in mice in diabetes research. This is mainly because there have been no clear reference values for mouse HbA1c, and reports on the consistency of measurements across different devices have been limited. However, recent studies in diabetic mice confirm that higher blood glucose levels correlate directly with higher HbA1c, indicating that it is a reliable measure of blood sugar in mice, much like it is in humans [146] (Figure 4, Table 1).

## 7. Case Studies of Biomarker Application in Serious Animal Diseases

Positive acute phase proteins (APPs) are expected to increase during inflammatory conditions [147]. Fibrinogen (Fb), haptoglobin (Hp), and serum amyloid A (SAA) are all positive APPs that are commonly studied in cattle with BRDC. Haptoglobin (Hp), an α-2 globulin, exhibits a bacteriostatic effect through its ability to bind hemoglobin [148]. In an experimental transmission trial, haptoglobin (Hp) levels showed a significant increase (*p* < 0.05) on day 4 following Mannheimia haemolytica (MH) inoculation. This increase was not observed with Bovine Herpesvirus 1 (BoHV1) inoculation on day 0. In a separate experimental trial, a significant elevation in Hp concentration (*p* < 0.001) was detected on the first day of MH challenge, which occurred four days after BoHV1 inoculation [149], which indicates the utility of Hp as an early biomarker of BRDS, especially for bacterial infections.

A 2025 study explored alterations in the serum amino acid profiles of dogs afflicted with intrahepatic and extrahepatic Congenital Portosystemic Shunts (CPSSs), comparing them to healthy control dogs. Employing a specialized amino acid analyzer, a form of metabolomic technology, the research revealed distinct perturbations in the amino acid profiles of CPSS-affected dogs, clearly differentiating them from their healthy counterparts. Specifically, the study identified increased serum concentrations of ammonia, asparagine, glutamic acid, histidine, phenylalanine, serine, and tyrosine. Conversely, decreased concentrations were noted for isoleucine, leucine, threonine, urea, and valine. The findings suggest that the post-surgical assessment of the serum amino acid concentration could serve as a valuable, non-invasive method to monitor the recovery of liver function in these animals [150].

Various biomarkers and their applications in detecting serious animal diseases. A single-stranded DNA aptamer was developed to specifically target the P48 protein of Mycoplasma Bovis, a significant pathogen causing bovine mastitis. This aptamer was utilized in a competitive enzyme-linked aptamer assay, demonstrating high sensitivity and selectivity comparable to commercial ELISA kits. In veterinary diagnosis, this method shows the potential of aptamer-based biosensors, allowing for the early detection of M. Bovis in blood serum samples [31]. This is a quick method for the detection of Ibaraki virus, which causes epizootic hemorrhagic disease. To identify the virus at picomolar concentrations, this method employs synchronous detection techniques and magnetic modulation. Fluorescent-labeled oligonucleotide works as a biomarker in this detection strategy, which allows the integration in portable devices for field use. This methodology emphasizes the advancement in biosensing technologies that facilitate the rapid and precise diagnosis of viruses in livestock [151]. For screening and monitoring cancer in dogs, an innovative technique has been developed, which focuses on measuring serum concentrations of various proteins.

A study that utilizes optical techniques states that 70% of dogs have abnormal levels of serum protein, diagnosed with cancer. This points out a significant correlation between changes in protein concentrations and the presence of cancer, suggesting the technique is a useful tool for identifying affected animals [152]. A study for the detection of lung injury used female WAG/RijCmcr rats, which had a good response to radiation. To observe the effects on the lungs, injured rats were exposed to different doses of whole thorax irradiation (WTLI), particularly 10 Gy and 13 Gy. For measuring lung perfusion, technetium-labeled macroaggregated albumin (99m Tc-MAA) and SPECT imaging techniques were used. A decrease in lung perfusion was observed at two weeks post-irradiation, indicating potential lung injury. Blood samples were analyzed for the total and differential white blood cell count. The study found that total white blood cells decreased after radiation exposure, which could indicate injury. Specific microRNAs (miRNAs) were measured in blood. Changes in the levels of certain miRNAs were associated with radiation exposure, providing another potential biomarker for lung injury [153].

There is a case study through a simulation of TAD control effort involving wild pigs (Sus scrofa), specifically in response to a hypothetical outbreak of African swine fever (ASF). The study simulated a control effort for TAD after detecting an index case in wild pigs. This approach is crucial as it allows researchers to explore operational challenges without the risks associated with real outbreaks. Three different removal methods were tested during the simulation. Aerial control involves using aircraft to manage the wild pig population from the air. Trapping is a traditional method that uses traps to capture wild pigs. Experimental Toxic Bait is an innovative method that tests the use of Bait laced with toxins to reduce pig populations. An after-action assessment highlighted several operational challenges faced during the simulation, such as the difficulty in coordinating rapid response efforts: ensuring the effectiveness of removal methods under various conditions; and managing the potential backlash against control measures and public perception. The simulation also added the recovery of carcasses from dispatched pigs. This is very crucial for managing diseases that spread through carcass-based transmission. For the prevention of the further spread of diseases, this aspect is very important.

To improve the TAD response operations, there is a need for strategic planning and technology advancement. This consisted of evolving better tools for controlling and monitoring populations of wildlife. Overall, the awareness gained from this simulation can ensure personnel are better equipped to handle real-life scenarios and can help build emergency response preparedness for future TAD outbreaks [154]. Rapid disease identification in livestock is essential for outbreak prevention. In livestock diseases like mastitis, biomarkers facilitate early detection in dairy cows, which will further reduce milk production if there are interventions. Many proteomic approaches have introduced new biomarkers for the detection of such diseases at the initial stages [155]. Control strategies include adjusting feeding practices, monitoring animal health in real time, and applying treatments that can be implemented by the farmers using biomarkers, for example, managing animal welfare and preventing disease outbreaks by identifying stress-related biomarkers in saliva [156].

Farmers achieve substantial economic savings through rapid disease detection and management using biomarkers. Similarly, farmers can maintain productivity and reduce losses associated with animal health issues by preventing severe outbreaks [157]. In controlling these diseases, the key strategy is to identify animals and exhibit resilience [158]. Recognizing the biomarkers related to resilience and strength of animals can lead to Disease Control and better management practices. To determine how these markers can be used to predict disease resistance, researching the mechanism of action of these biomarkers is involved. We can develop new strategies to control the spread of mycobacterial diseases by increasing our understanding of resilience-related biomarkers. This could involve implementing management practices that support the health of livestock or selective breeding for resilient traits [159].

## 8. Challenges and Future Perspectives

There are several limitations in the detection of animal diseases through biomarkers. These challenges are due to the complexities of the development of biomarkers and challenges associated with current technologies. Many biomarkers remain in the beginning stages and have not transitioned to clinical practice, regardless of the advancements in ‘omics’ technologies. Due to the insufficient validation and compatibility with clinical needs, any biomarker identified through research does not reach clinical application [40,160]. Measurement fidelity represents a fundamental limitation in current biomarker detection technologies. This indicates the precision and consistency of the sensor’s readings. As sensors become capable of detecting lower concentrations of target molecules, they also try to respond to more interfering substances. This can lead to increased false-positive diagnoses, which challenge the reliability of the diagnosis [161]. Financial support, the limited sample size, difficulties in sample collections, species variation, the lack of standardization, and insufficient bioinformatics resources for analyzing data from new technologies in veterinary medicine are the key challenges in the detection of animal disease through biomarkers [40]. Similarly, degradation affecting test sensitivity is also an obstacle in disease diagnosis through biomarkers. As seen in the study, we have 60 specimens’ letters omitted due to extreme hemolysis, highlighting the difficulty in obtaining usable specimens from affected animals [162].

Even though we have made promising strides in finding new biomarkers for veterinary use, it has been difficult to turn these discoveries into useful diagnostic tools because of the significant regulatory and standardization hurdles. A major obstacle is the absence of standardized tests that can be used across different animal species. Many of the tests for biomarkers, particularly those for proteins and peptides, need reagents and calibration specifically for one species, which makes it hard to scale them up for widespread use among different livestock [163]. There are not too many global processes for getting veterinary diagnostic biomarkers approved. For example, in the United States, the USDA Center for Veterinary Biologics is in charge of approving diagnostic kits, and they require proof that the kits are safe, effective, and work well. However, there is no official system in place to approve more complex molecular biomarkers, such as those used in proteomics or metabolomics [164]. In Europe, the rules for veterinary diagnostics, which are a type of in vitro diagnostic (IVD), are not consistently applied. Because each country handles this differently, it makes the process of commercializing new biomarkers fragmented and inconsistent across the continent [41]. Furthermore, transitioning biomarkers from bench to barn requires not just regulatory clearance, but technological readiness and market translation strategies. Many biomarkers identified through ‘omics’ research fail to reach point-of-care implementation due to the high costs, lack of portable platforms, and limited collaborations between researchers and diagnostic companies [32].

The diversity of farm animal species complicates proteome analysis and interpretation, and the presence of albumen-like high-abundance proteins in serum complicates the identification of specific biomarkers [155]. The complex multifactorial nature of diseases presents the primary challenge for biomarker identification regarding the indication of specific and sensitive biomarkers from different molecules present in tissues and biological fluids [16]. Enabling rapid and cost-effective biomarker discovery and monitoring, surface-enhanced Raman spectroscopy (SERS) is a technology that advances a high sensitivity and label-free operations. SERS has notable abilities to provide accurate monitoring from biological sources such as cells, tissues, and biofluids [165]. The single-molecule array (SIMOA) is an ultra-sensitive detection method that has shown potential in the detection of blood-based biomarker diseases such as Alzheimer’s, enabling early detection via easily obtained samples [166]. Utilizing various nanomaterials for a rapid and selective analysis that are important for a liquid biopsy, nanomaterial-based microfluidics enhances the detection of biomarkers for cancer [167]. Integrating biological and physical signs to increase diagnostic precision, wearable and implantable sensors aim to consistently monitor biomarkers. For detecting circulating tumor cells and DNA, and body fluids, liquid biopsy is an emerging method: it can simplify cancer diagnostics and reduce the need for invasive procedures [168].

## 9. Practical Constraints in Resource-Limited Settings

### 9.1. Artificial Intelligence Implementation Barriers

While AI offers transformative potential for disease detection, its real-world application faces significant hurdles in typical veterinary environments. The high infrastructure costs particularly for GPU-dependent imaging analysis remain prohibitive for 89% of rural clinics in developing regions [23]. More critically, AI models trained predominantly on data from commercial European dairy farms show severe performance degradation when applied to indigenous cattle breeds, with mastitis detection accuracy dropping from 92% to 64% in field validations [25]. Even when technical solutions exist, the limited digital literacy among veterinarians and farmers creates adoption barriers; mobile-optimized tools like lameness detection apps require comprehensive training programs that are unavailable in 74% of low-income regions [32].

### 9.2. CRISPR Diagnostic Challenge

The promising sensitivity of CRISPR-based diagnostics (e.g., 10 copies/μL detection of CPV) is counterbalanced by practical constraints in veterinary field settings [30]. No USDA-approved veterinary CRISPR kits currently exist, and the technology’s strict cold-chain requirements for reagents are economically unsustainable in tropical regions without reliable refrigeration [1]. Biological complexities further limit validity: in salmonids and other polyploid species, off-target effects generate false positives in 22% of cases [30]. While paper-based lateral flow formats (e.g., FELUDA) reduce costs by 90%, their inability to multiplex pathogens restricts the clinical utility for syndromic diseases like BRDC [1].

### 9.3. Liquid Biopsy Viability Gaps

Liquid biopsies face fundamental limitations in resource-limited contexts. The instability of circulating biomarkers requires immediate plasma separation at −80 °C, a condition unmet in 92% of field clinics [17]. When compromised storage occurs, hemolysis in bovine samples increases false negatives by 3.7-fold [28]. More fundamentally, critical validation gaps persist: breed-specific reference ranges for circulating tumor DNA remain undefined even in high-value species like dairy cattle (e.g., Gir vs. Holstein thresholds) [17]. Emerging microfluidic concentration systems (CellRaft AIR^®^) enable barn-side processing but currently detect only high-abundance biomarkers, missing early disease states [167].

## 10. Conclusions

This review synthesizes critical advances in veterinary biomarker science, highlighting its indispensable role in early disease detection for conditions with significant economic and public health impacts, such as antibiotic resistance. Our analysis reveals three core insights: First, biomarker mechanisms are now well-defined across species, with diagnostic tools like CRP in canine sepsis, prognostic indicators such as cfDNA in oncology, and predictive signatures like BRAF in melanoma enabling more targeted interventions. Second, the technological readiness of these tools varies significantly: while technologies like AI hematology analyzers are highly validated (with 98.7% sensitivity), others, such as LC-MS/MS metabolomics, are in a transitional phase due to cost limitations, and CRISPR diagnostics remain experimental with no field deployment. Finally, our review offers practical lessons, underscoring that point-of-care adoption requires extensive species-specific validation, microfluidic biosensors show near-term promise for resource-limited settings at a low cost, and multi-omics integration is essential for understanding and managing complex diseases like BRDC.

## Figures and Tables

**Figure 1 animals-15-03132-f001:**
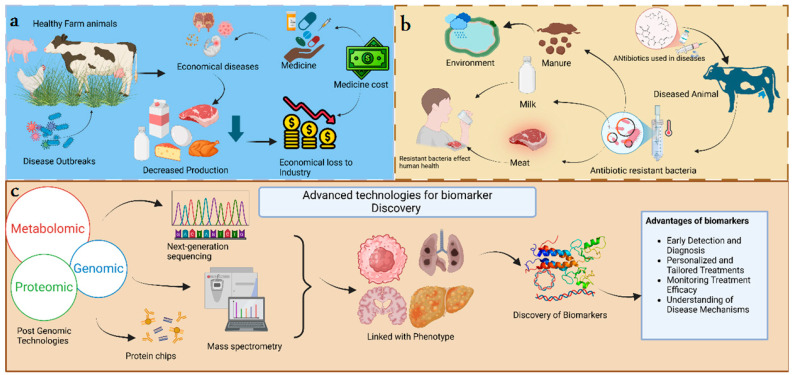
The graphical representation of introduction (**a**) explains the economic losses related to animal diseases. When disease outbreaks attack healthy farm animals, they cause economic losses in medicinal costs and decrease production, affecting the revenue from the products [1,2,3]. (**b**) Public health is at risk. When there is nontherapeutic use of antibiotics in diseased animals, it promotes the development of resistant bacteria in the animal body that are transferred to the public when they consume the contaminated products from the diseased animals. Livestock promotes the development of resistant bacteria [3]. (**c**) Advanced technologies like post-genomic technologies are linked with phenotypes of traits associated with diseases. This process promotes the discovery of biomarkers, which has vital advantages [4].

**Figure 2 animals-15-03132-f002:**
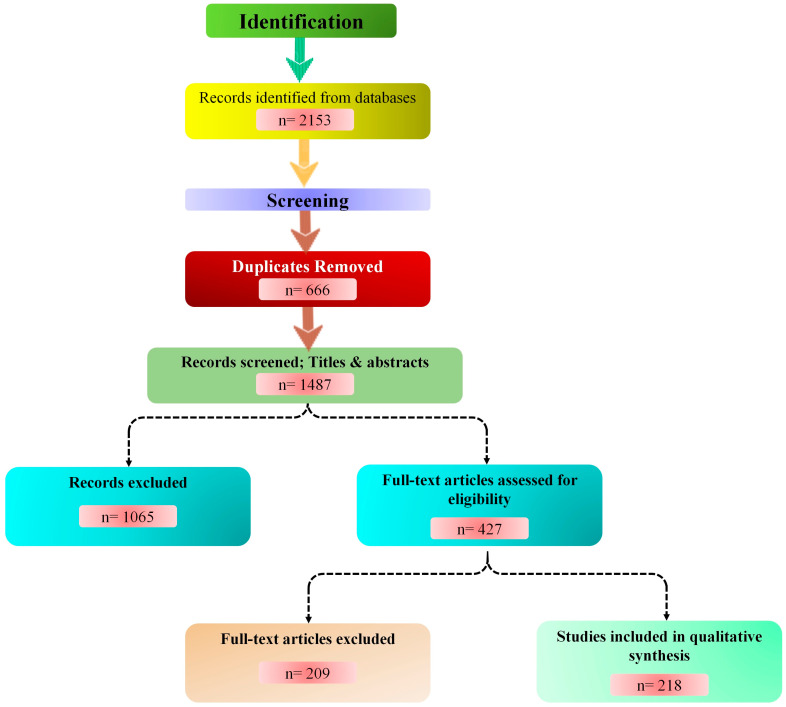
PRISMA flow diagram. Schematic of the literature search and screening process. Databases included PubMed, Web of Science, Scopus, CAB Abstracts, IEEE Xplore, and Google Scholar. Out of 2153 initial records, 218 studies met the inclusion criteria. Figure created with the help of Mermaid Chart (https://www.mermaidchart.com).

**Figure 3 animals-15-03132-f003:**
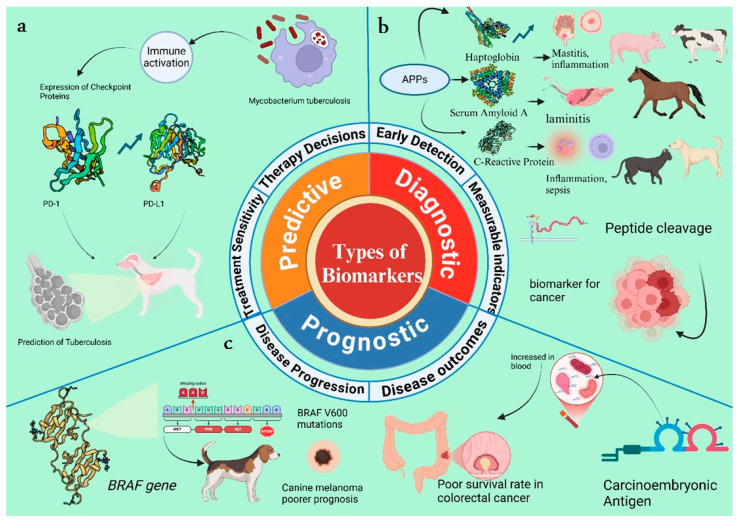
Types of biomarkers: (**a**) Predictive biomarkers are helpful in therapy decisions and in checking the treatment sensitivity, and whether the treatment is effective or not. For example, in canine tuberculosis, infection initiates an immune regulatory mechanism that upregulates checkpoint proteins like PD-L1 in immune cells. Elevated levels of PD-1 expression may indicate a stronger immunosuppressive response and can predict the response to therapies [2,72]. (**b**) Diagnostic biomarkers are measurable indicators that detect the disease at its earliest stage. For example, acute-phase proteins (APPs) are widely used as sensitive indicators of systemic inflammation, infection, and tissue injury in animals. Increased levels of haptoglobin (Hp) indicate inflammatory responses in cattle and pigs. Serum amyloid A is widely used to diagnose respiratory infections and laminitis in horses. Similarly, C-reactive protein indicates inflammation and sepsis in cats and dogs. Peptide cleavage is a biomarker of cancer [73]. (**c**) Prognostic biomarkers indicate the progression and outcomes of the disease. For example, the BRAF v600 mutation in the BRAF gene is an indicator of poorer prognosis in canine melanoma. Increased carcinoembryonic antigen (CEA) levels in the blood are used for the prognosis of larger tumors in colorectal cancer [74].

**Figure 4 animals-15-03132-f004:**
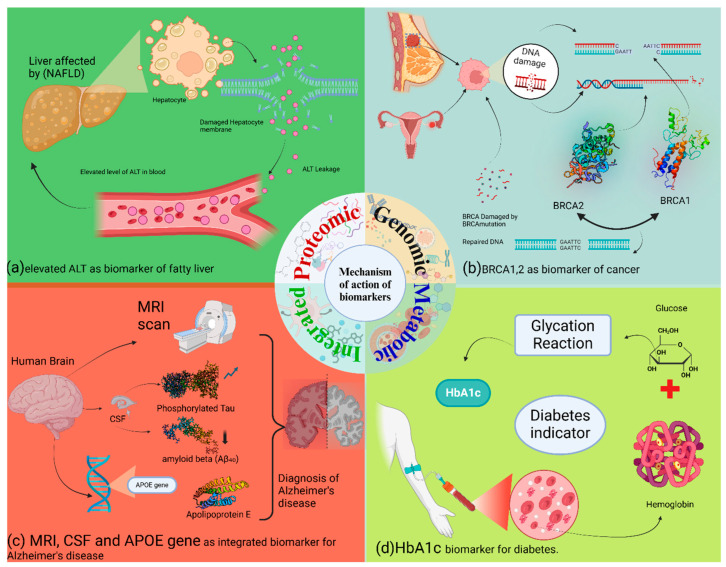
Explain the mechanism of action of different biomarkers. (**a**) In non-alcoholic fatty liver disease (NAFLD), hepatocyte membrane compromise leads to the release of alanine aminotransferase (ALT) into the bloodstream. Elevated serum ALT serves as a biomarker for liver damage. (**b**) BRCA1/BRCA2: These genes encode proteins critical for DNA double-strand break repair via homologous recombination. Loss-of-function mutations in *BRCA1* or *BRCA2* impair this repair pathway, leading to genomic instability and increased cancer risk. (**c**) Integrated AD biomarkers: Alzheimer’s disease diagnosis employs an integrated approach. MRI detects structural brain changes. Cerebrospinal fluid (CSF) analysis measures altered levels of amyloid-beta and phosphorylated tau proteins. Genetic testing identifies risk alleles, such as in the *APOE* gene. (**d**) HbA1c: Hemoglobin A1c is formed through the non-enzymatic glycation of hemoglobin in red blood cells. Its concentration reflects average blood glucose levels over the preceding 2–3 months, making it a stable biomarker for long-term glycemic control in diabetes.

**Table 2 animals-15-03132-t002:** Recent advances in genomic technologies for animal disease biomarker discovery.

Technology/Method	Key Principle	Applications in Animal Diseases	Recent Examples			Source
**NGS**	Simultaneous reading of millions of DNA fragments for rapid, cost-effective genome sequencing; platforms include Illumina, PacBio, Ion Torrent; bioinformatics tools assemble reads into complete genomes	Comprehensive genomic profiling, viral research (mutation tracking, vaccine development), rare genetic conditions diagnosis	**Canine Rare Genetic Disorders:** Diagnosis of suspected genetic disorders in pediatric patients identifying novel variants (35.9%)	**Viral Diseases:** Tracking mutations in Foot-and-Mouth Disease Virus (FMDV), and monitoring Avian Influenza, and African Swine Fever Virus (ASFV) for vaccine effectiveness and outbreak control		[89,90]
**RNA Sequencing (RNA-Seq)**	Analysis of entire RNA molecules (transcripts) to provide insights into gene expression, alternative splicing, and regulatory mechanisms; RNA extracted, converted to cDNA, then sequenced by NGS	Gene expression profiling, understanding disease progression, identifying therapeutic targets, rare disease diagnosis, drug repurposing	**Canine Invasive Urothelial Carcinoma (iUC):** Identified 2531 differentially expressed genes; downregulation of TP53, upregulation of ERBB2; mutations in FGFR3; increased PD-L1 expression	**Canine Melanoma:** Downregulation of MAPK and PI3K/AKT pathways; upregulation of NOS2; overexpression of miR-450b leading to increased MMP9 expression	**Canine Osteosarcoma (OS):** Single-cell RNA-Seq revealed 41 distinct cell types, including novel tumor cell clusters with interferon response gene signatures and specific mregDCs; high cross-species similarity with human OS	[90,91]
**Epigenomics (DNA Methylation, Histone Modifications)**	Study of heritable changes in gene function without DNA sequence alteration; involves marks like DNA methylation and histone modifications; analyzed by ChIP-seq (protein-DNA interactions) and ATAC-seq (chromatin accessibility)	Animal health and welfare monitoring, disease resistance, origin tracing, aging research, breeding programs	**Broiler Chickens:** DNA methylation clock showed accelerated aging with induced systemic inflammation (2023), predicting health/performance	**Livestock/Aquaculture:** Location-specific DNA methylation signatures identified in shrimp, salmon, and chickens for origin tracing and assessing practices like antibiotic usage	**Mice (Aging):** Breakdown in epigenetic information drives aging, restoration reverses signs of aging; increased aging biomarkers with epigenetic disorganization	[92]
**Single-Cell Genomics (scRNA-seq, scATAC-seq)**	Analysis of genetic sequences at individual cell level to resolve cellular heterogeneity; scRNA-seq for gene expression, and scATAC-seq for chromatin accessibility	Uncovering rare cell populations, understanding cellular differentiation/lineage, high-resolution disease insights, biomarker development	**Canine Osteosarcoma (OS):** Revealed 41 distinct cell types in TME, including novel tumor cell and immune cell populations; identified transcriptional heterogeneity within malignant osteoblasts	**Chickens (Pimpled Eggs):** Integrated scRNA-seq and scATAC-seq identified ionocytes, TFs (ATF3, ATF4, JUN, FOS), regulating uterine activity, and ion pump downregulation linked to egg formation	**Bovine Genomics:** Comprehensive catalog of cis-regulatory elements (CREs) in cattle using scATAC-seq (2023); insights into chromatin accessibility in oocytes/embryos and muscle growth in Tianzhu	[89]

## Data Availability

No data was used for the research described in the article.

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
