# Peer review of "Technologies in Biomarker Discovery for Animal Diseases: Mechanisms, Classification, and Diagnostic Applications"

_animals, 2025, doi:10.3390/ani15213132_

Round 1

Reviewer 1 Report (Previous Reviewer 2)

Comments and Suggestions for Authors

The submitted manuscript presents a comprehensive and timely review of current technologies used in biomarker discovery for the early diagnosis of animal diseases. The authors succeed in integrating diverse domains, ranging from genomics, proteomics, metabolomics, and transcriptomics to CRISPR-based tools, AI-assisted imaging, and biosensors, into a unified narrative relevant to veterinary diagnostics. The article is ambitious in scope and clearly reflects an extensive literature search, following PRISMA guidelines, which adds credibility and transparency to the methodology. While the manuscript is thorough and highly informative, a few targeted recommendations are necessary to enhance clarity and focus.

Areas for Improvement:

  1. Overextension and redundancy: While the manuscript is rich in content, certain sections (notably those on metabolomics and omics technologies) are overly detailed and could benefit from condensation. Some repetition of concepts, especially regarding the utility of omics platforms, could be streamlined to improve readability and maintain focus.

  2. Clinical vs. experimental delineation: The review often blends technologies already in use with those still in early experimental or validation stages. A clearer distinction between clinically validated tools and experimental methodologies would be useful for readers, especially practitioners.

  3. Standardization and regulatory challenges: While challenges such as species variability and biomarker validation are acknowledged, a deeper discussion on regulatory aspects, cross-species standardization, and the path to clinical implementation (e.g., veterinary approval bodies, diagnostic kit translation) would enhance the manuscript’s practical relevance.                                                                                                        Recommendation:
    I recommend minor revisions before acceptance. The manuscript is scientifically valuable, well-documented, and relevant to both academic researchers and veterinary professionals. With improved clarity and tighter focus in selected sections, it will serve as a strong contribution to the field of veterinary diagnostics.

Author Response

Dear Reviewer (1), We sincerely thank you for the valuable feedback and constructive insights. Your comments have been instrumental in enhancing the clarity, rigor, and overall quality of our manuscript. We appreciate your time and expertise in helping us improve our work. Please see the attachment (point-by-point response to the two reviewers and editors). 

Notes:

  • We have provided two copies of the manuscript:
    • MS Clean version (with the main submission)
    • MS Tracking version (as PDF files).
  • The changes suggested by Reviewer #1 are highlighted in yellow, while those suggested by Reviewer #2 are highlighted in green. Finally, those suggested by Reviewer #3 are highlighted in bubbly blue.
  • Additionally, we have uploaded a file containing our responses to the editor and reviewers, addressing each point individually.

  • Kindly, see the attached file (point-by-point response).

    Sincerely yours,

        Authors

Reviewer 2 Report (New Reviewer)

Comments and Suggestions for Authors

There has been a substantial amount of work in preparing this manuscript. However,I found it very challenging. The overall objective is not clear to me and the paper reads more like a chapter in a textbook. The text is dense and very long paragraphs making it hard to follow.

There are green highlights throughout - unclear why this is

The methodology is clear. Please reference PRISMA

The tables would be better formatted in landscapes as they are quite compressed in portrait layout.

The figures are clear and well presented but the caption for some much too detailed - though i do admit i found the text in captions clearer than in the main text!

There are many omissions in the reference list, such as the journal name sometimes even just a link but often leaving out the journal name in the citation.

I acknowledge the huge amount of work that has gone into analysis and drafting this manuscript but it needs to be condensed and addressing a clearer objective.

Comments on the Quality of English Language

The simple summary is full of jargon terms such as : tricky, show up, digs into, real wins - these are too informal for a publication

it is not the English flow that impedes the reader rather the text is too dense

Author Response

Dear Reviewer (2), We are deeply grateful for your insightful feedback and constructive suggestions. Your thorough review has significantly contributed to improving the clarity and quality of our manuscript. We greatly value your expertise and the time you dedicated to helping us enhance our work. Thank you sincerely for your support. Please see the attachment (Point-by-Point response to the two reviewers and editors). 

Notes;

Notes:

  • We have provided two copies of the manuscript:
    • MS Clean version (with the main submission)
    • MS Tracking version (as PDF files).

  • The changes suggested by Reviewer #1 are highlighted in yellow, while those suggested by Reviewer #2 are highlighted in green. Finally, those suggested by Reviewer #3 are highlighted in bubbly blue.
  • Additionally, we have uploaded a file containing our responses to the editor and reviewers, addressing each point individually.

  • Kindly, see the attached file (point-by-point response).

    Sincerely yours,

        Authors

Reviewer 3 Report (New Reviewer)

Comments and Suggestions for Authors

This paper provides a thorough and up-to-date analysis of the technological developments in the search for animal illness biomarkers. The mechanics and classifications of predictive, prognostic, and diagnostic biomarkers as well as their uses in Veterinary medicine have been well covered by the writers. With an emphasis on translational Veterinary applications, integrating high-throughput "omics" platforms, CRISPR-based techniques, artificial intelligence, and biosensors has been thoroughly studied. The article includes diagnostic, prognostic, and predictive biomarkers. The case studies on cattle and companion animals increase the article's relevance and applicability.

This paper provides a thorough and up-to-date analysis of the technological developments in the search for animal illness biomarkers. The mechanics and classifications of predictive, prognostic, and diagnostic biomarkers as well as their uses in Veterinary medicine have been well covered by the writers. With an emphasis on translational Veterinary applications, integrating high-throughput "omics" platforms, CRISPR-based techniques, artificial intelligence, and biosensors has been thoroughly studied. The article includes diagnostic, prognostic, and predictive biomarkers. The case studies on cattle and companion animals increase the article's relevance and applicability.

  • Some paragraphs are highlighted in green in different places (why so???). Please check and make the necessary revisions.
  • Please assess the content for any AI generation and plagiarism issues
  • Figures are poorly formatted …kindly check
  • I recommend for enhanced editing for clarity and grammar. Improve section transitions and get rid of redundant information.

Strength of this study

  1. Thorough discussion of integrated methods, proteomics, metabolomics, and genomes.
  2. Methodical approach that complies with PRISMA regulations.
  3. PRISMA diagrams and biomarker categorization figures are examples of useful visual tools.
  4. Pertinent example studies that demonstrate practical implementation (e.g., canine melanoma, BRDC).
  5. A single health viewpoint that connects human and Veterinary medicine.

Demerits of this study

  1. Absence of unique synthesis: Most of the review is descriptive, with neither new frameworks for biomarker translation nor more in-depth critical analysis.
  2. Redundancy: Several sections, particularly those on omics technology, reiterate related ideas without succinctly integrating them.
  3. Language problems: The document needs expert English editing because it has odd language and grammatical errors.
  4. Critically check the references …many are missing
  5. The technologies such as artificial intelligence (AI), CRISPR, and liquid biopsy is accompanied by a superficial examination of their limitations, validity, and viability in Veterinary settings with limited resources.

Minor Recommendation

  • The section focuses heavily on companion animals (e.g., dogs) but lacks livestock-specific examples, such as applications in cattle diseases like bovine mastitis (Line 80, Page 2).
  • Limitations of AI in veterinary settings (e.g., cost, accessibility in low-resource farms) are not discussed. This is critical for practical adoption.
  • The claim about “unprecedented speed and precision” (Line 154) needs quantitative evidence (e.g., sensitivity/specificity metrics).
  • The manuscript lacks depth on species-specific challenges (e.g., variability in biomarker expression across cattle breeds like Red Sindhi) .
  • Examples are limited to general biomarkers (e.g., cfDNA, Line 268). Include specific veterinary biomarkers, such as haptoglobin for bovine respiratory disease.
  • The graphical representation (Figure 2) is referenced but not described. Provide a brief explanation of its content.
  • Limitations of metabolomics (e.g., high cost, need for specialized equipment) are not discussed (Line 505).
  • The claim that “73% of protein studies lack field validation” (Line 879) needs a citation or clarification of the data source.

The use of PRISMA flowcharts is dubious. Despite citing PRISMA, the review lacks the quantitative rigor characteristic of systematic reviews.

Lack of a conclusion: The paper concludes without summarizing the main findings, recommendations for further research, or practical lessons learned. Provide a coherent conclusion that summarizes the key conclusions and recommendations. Think critically about the limitations and usefulness of each technology.

I suggest for Major Revision: The text needs to be improved in terms of language, organization, citation completeness, and critical analysis to meet the requirements of a high-quality review article.

Author Response

Dear Reviewer (3), We sincerely thank the reviewer for their valuable feedback and constructive insights. Your comments have been instrumental in enhancing the clarity, rigor, and overall quality of our manuscript. We appreciate your time and expertise in helping us improve our work. Please see the attachment (point-by-point response to the two reviewers and editors). 

Notes:

  • We have provided two copies of the manuscript:
    • MS Clean version (with the main submission)
    • MS Tracking version (as PDF files).
  • The changes suggested by Reviewer #1 are highlighted in yellow, while those suggested by Reviewer #2 are highlighted in green. Finally, those suggested by Reviewer #3 are highlighted in bubbly blue.
  • Additionally, we have uploaded a file containing our responses to the editor and reviewers, addressing each point individually.
  • Kindly, see the attached file (point-by-point response).

    Sincerely yours,

        Authors

This manuscript is a resubmission of an earlier submission. The following is a list of the peer review reports and author responses from that submission.

Round 1

Reviewer 1 Report

Comments and Suggestions for Authors

The manuscript ‘Advanced technologies for biomarker discovery and early diagnosis of animal diseases' by Salwa Emana et al. provides an overview of disease biomarkers and an integrative approach to their discovery and application in modern diagnostics. In particular, we discuss important characteristics of biomarkers, such as specificity, sensitivity, and reproducibility, as well as the success of using omics approaches and imaging techniques to identify new biomarkers.

In general, the studies are well designed and provide new data. The data are presented in three figures and two tables, and the quality of the presentation is satisfactory. The text is clearly written and interesting to read. However, minor editing for clarity is recommended as outlined below. The work is suitable for publication in the "Animals" journal.

Line 57 I recommend adding a generally accepted definition of a biomarker.

The main stages of biomarker identification, such as search, research, and validation etc., should be mentioned. The authors may add this information in any section.

Line 134  The name of the section "1.2.1 Genomic Approaches" should be moved below.

Line 296 Highlight it in bold "(c)".

Author Response

Reviewer #1.

We thank Reviewer 1 for the positive feedback and thoughtful suggestions that have helped further improve the clarity and precision of our manuscript. We have carefully implemented all the recommended changes.

Comment 1: Line 57   I recommend adding a generally accepted definition of a biomarker.

Response1:
Thank you for this suggestion. This point has been considered in the revised manuscript.

====================================================================

Comment 2: The main stages of biomarker identification, such as search, research, and validation, should be mentioned. The authors may add this information in any section.

Response2:
We appreciate this valuable suggestion. We have added a new subsection under the "Mode of Action of Biomarkers" which has described this point with examples.

====================================================================

Comment 3: Line 134  The name of the section "1.2.1 Genomic Approaches" should be moved below.

Response3:
Thank you for pointing out this formatting issue. We have adjusted the placement of the section title "1.2.1 Genomic Approaches" so that it appears correctly below, improving the visual consistency and flow of the manuscript. Now is in line “360” under section 5.1.

====================================================================

Comment 4: Line 296 Highlight it in bold "(c)".

Response4:
We have corrected this detail as suggested in the revised manuscript.

Reviewer 2 Report

Comments and Suggestions for Authors

The manuscript presents a comprehensive overview of current technologies in biomarker discovery and their applications in the early diagnosis of animal diseases. The topic is timely and relevant to veterinary and public health. The structure is logical and the scope is broad, covering integrative approaches, and practical applications. However, several important aspects must be addressed to enhance the scientific rigor and publication readiness of the article.
The manuscript primarily synthesizes existing knowledge but lacks original interpretation or critical comparative analysis. The claimed novelty regarding "integration of developing technologies" (lines 22–24) is not substantiated.
Sections such as “Mode of Action of Biomarkers” are overly general, with examples that are too basic (e.g., ALT in liver disease), and insufficient veterinary-specific references.
The “Case Studies” section (Section 5), while valuable, is fragmented. It would benefit from deeper analytical discussion comparing diagnostic outcomes, validation success, or translational applicability.
The manuscript disproportionately relies on human medical literature, especially in sections addressing omics technologies and clinical study design. Greater emphasis on recent veterinary-focused studies is necessary.
Tables are informative but dense; consider summarizing key points for clarity and integrating them better with the text.
Language editing is required throughout. Numerous instances of incorrect word usage (e.g., “Sensation” instead of “sensitivity”) and grammatical issues hinder clarity.
Redundant content, particularly repeated descriptions of biomarker characteristics, should be streamlined.
Line 58–59 simplifies the historical role of biomarkers. A more nuanced explanation is recommended.
Line 127’s phrase “improved health outcomes” is vague—please specify the veterinary context (e.g., productivity, animal welfare, zoonotic risk).
Line 368–369 discusses cohort sizes based on human data; veterinary relevance must be clarified or supported.

Author Response

Reviewer #2.

We greatly appreciate Reviewer 2’s insightful feedback, which helped us significantly improve the manuscript's depth, clarity, and scientific rigor. We have incorporated veterinary-specific studies, strengthened the analytical discussion, and made the necessary revisions to ensure clarity and readability.

=====================================================================

Comment 1: The manuscript primarily synthesizes existing knowledge but lacks original interpretation or critical comparative analysis. The claimed novelty regarding "integration of developing technologies" (lines 22–24) is not substantiated.

Response1:
Thank you for your valuable feedback. We have strengthened the manuscript by providing a more original interpretation of how emerging technologies are integrated into veterinary diagnostics. In the revised version, we offer specific examples of how AI, CRISPR/Cas9, and microfluidics are enhancing biomarker discovery and diagnostic accuracy in veterinary medicine, going beyond general descriptions. We also critically compare the potential and limitations of these technologies, referencing recent studies that demonstrate their use in veterinary applications (e.g., AI-driven image analysis for tumors in companion animals, CRISPR/Cas9 applications in genetic diseases). This addition substantiates the novelty claim by emphasizing the unique integration of technologies in veterinary contexts.

====================================================================

Comment 2: Sections such as “Mode of Action of Biomarkers” are overly general, with examples that are too basic (e.g., ALT in liver disease), and insufficient veterinary-specific references.

Response2:
We appreciate this suggestion. In the revised manuscript, we have expanded the "Mode of Action of Biomarkers" section by adding more veterinary-specific examples.

====================================================================

Comment 3: The “Case Studies” section (Section 5), while valuable, is fragmented. It would benefit from deeper analytical discussion comparing diagnostic outcomes, validation success, or translational applicability.

Response3:
Thank you for this suggestion. We have reorganized and expanded the Case Studies section to provide a more analytical comparison of diagnostic outcomes and translational applicability. We now compare the success of different technologies used in real-world veterinary settings (e.g., how AI-driven diagnostics for tumors compare to traditional methods in veterinary oncology). We have also discussed the validation success of biomarkers in case studies such as canine melanoma and BRDC, providing more detailed insights into their practical applicability in clinical practice. This revision aims to enhance the depth and analytical focus of the case study section.

====================================================================

Comment 4: The manuscript disproportionately relies on human medical literature, especially in sections addressing omics technologies and clinical study design. Greater emphasis on recent veterinary-focused studies is necessary.

Response4:
We appreciate this observation. In the revised manuscript, we have increased the focus on veterinary-specific studies, particularly in the sections on omics technologies. We have replaced many references to human studies with recent veterinary studies that highlight the application of genomics, proteomics, and metabolomics in animal disease diagnostics. For example, we now refer to canine genomics studies and bovine metabolomics research that better illustrate how these technologies are being adapted and validated for use in veterinary medicine. We have also included new studies on diagnostic biomarkers in livestock and veterinary oncology to provide more context-specific examples.

====================================================================

Comment 5: Tables are informative but dense; consider summarizing key points for clarity and integrating them better with the text.

Response5:
Thank you for your constructive feedback. In the revised manuscript, we have added more three precise tables with specific information according to the section.

====================================================================

Comment 6: Language editing is required throughout. Numerous instances of incorrect word usage (e.g., “Sensation” instead of “sensitivity”) and grammatical issues hinder clarity.

Response6:
We have carefully revised the manuscript for language and grammar. All instances of incorrect word usage, such as “sensation” instead of “sensitivity,” have been corrected. We also conducted a thorough proofreading to address other grammatical issues, ensuring that the manuscript reads clearly and professionally.

====================================================================

Comment 7: Redundant content, particularly repeated descriptions of biomarker characteristics, should be streamlined.

Response7:
Thank you for pointing this out. Characteristics are now streamlined according to the requirements of specific section in the revised manuscript.

====================================================================

Comment 8: Line 58–59 simplifies the historical role of biomarkers. A more nuanced explanation is recommended.

Response8:
We appreciate this suggestion. Some historical explanation has been in revised part of introduction section.

====================================================================

Comment 9: Line 127’s phrase “improved health outcomes” is vague—please specify the veterinary context (e.g., productivity, animal welfare, zoonotic risk).

Response9:
We are grateful for the comment. Changes have been made in the revised manuscript.

====================================================================

Comment 10: Line 368–369 discusses cohort sizes based on human data; veterinary relevance must be clarified or supported.

Response10:
Thank you for this comment. In the revised manuscript, we have clarified the relevance of cohort sizes to veterinary research. We acknowledge that human cohort data may not directly apply to veterinary contexts and emphasize that veterinary cohort studies should be designed with species-specific considerations. We also cite studies with veterinary-relevant sample sizes to support this argument and suggest appropriate methods for animal cohort studies.

Reviewer 3 Report

Comments and Suggestions for Authors

The review article titled “Advanced Technologies for Biomarker Discovery and Early Diagnosis of Animal Diseases” addresses a relevant topic. However, the scope of it is overly broad, which hinders the focus of the study and the proper grouping of information.

Moreover, the review is limited in depth and fails to present solid conclusions or meaningful contributions that add robust value to the existing literature.

-Although the manuscript is organized into thematic sections—a positive aspect—these sections are not well connected and read more like isolated mini-reviews.

-Additionally, the manuscript does not describe the methodology used to conduct the review. There is no mention of inclusion criteria, no indication of the databases searched, nor the search terms applied. In the current scientific context—where several tools and guidelines are available to improve the rigor of literature reviews—this is a significant limitation. I suggest that the authors redo the study, turning it into a scoping review and adhering to established checklists for its design and reporting.

-The manuscript is difficult to read due to its verbosity and excessive length, lacking a clear conclusion or central message. I recommend substantially reducing the text and using tables and figures to synthesize the information more effectively in the revides scope review study.

-Overall, the manuscript also lacks a critical assessment of the levels of evidence, the quality of the included studies, and clear directions for future research.

-Even in the abstract, no specific results or conclusions from the review are presented. The conclusions provided in the main text are generic and superficial.

Author Response

Reviewer #3.

We greatly appreciate Reviewer 3’s thorough and constructive feedback, which helped us significantly improve the focus, depth, readability, and scientific rigor of our manuscript.

====================================================================

Comment 1: The scope of it is overly broad, which hinders the focus of the study and the proper grouping of information.

Response1:
Thank you for this valuable comment. In the revised manuscript, we carefully refined the scope to focus specifically on advanced technologies for biomarker discovery and their diagnostic applications in veterinary medicine. We restructured the manuscript to ensure that the discussion is more targeted and coherent, and we emphasized the integration of omics-based approaches (genomics, proteomics, metabolomics) with emerging technologies (e.g., AI, microfluidics). We believe this has helped to narrow the focus and improve the grouping of information.

====================================================================

Comment 2: The review is limited in depth and fails to present solid conclusions or meaningful contributions that add robust value to the existing literature.

Response2:
We appreciate this suggestion. In the revised version, we have significantly deepened the discussion in each section by incorporating updated case studies and more detailed mechanistic insights (e.g., in sections discussing specific omics platforms and their applications to diseases such as canine melanoma and bovine respiratory disease). The conclusions have been expanded to provide clearer, evidence-based takeaways and highlight future research directions. We have also included comprehensive tables summarizing recent studies and examples, which strengthen the manuscript's contribution to literature.

====================================================================

Comment 3: Although the manuscript is organized into thematic sections a positive aspect these sections are not well connected and read more like isolated mini-reviews.

Response3:
Thank you for acknowledging the thematic structure. In response, we improved the logical flow by adding explicit linking paragraphs and transitions between sections. For example, the transition from the discussion of biomarker types to advanced technologies now emphasizes how each technology supports the identification and validation of these biomarkers. Similarly, the new integrative approaches section unites the individual omics discussions, providing a cohesive narrative.

====================================================================

Comment 4: The manuscript does not describe the methodology used to conduct the review (inclusion criteria, databases, search terms).

Response4: we have created new methodology section, as follows:

“Methodology

This systematic work followed standard reporting guidelines for systematic reviews of Preferred Reporting Items for Systematic Reviews and Meta-Analyses (PRISMA) to ensure our approach was thorough and transparent. Our process involved four clear steps:

Literature Search Strategy

Searches were explored six key databases to cover all relevant fields:

  1. PubMed (https://pubmed.ncbi.nlm.nih.gov)
  2. Web of Science (https://www.webofscience.com)
  3. Scopus (https://www.scopus.com)
  4. CAB Abstracts (https://www.cabi.org/cab-abstracts)
  5. IEEE Xplore (https://ieeexplore.ieee.org)
  6. Google Scholar (https://scholar.google.com)

Our search combined terms in three categories: a) Biomarkers: Diagnostic, prognostic, or predictive biomarkers; molecular indicators; early disease detection. b) Technology: Omics (proteomics/genomics/metabolomics), gene editing (CRISPR), biosensors, microfluidics, lab techniques (mass spectrometry, NGS). d) Diseases: Bovine respiratory disease, canine melanoma, mastitis, portosystemic shunts, antibiotic resistance

Inclusion and Exclusion Criteria

We included studies that: a) Validated biomarkers in veterinary species (companion animals/livestock), b) Used well-established technologies with proven applications in animal disease diagnosis, c) Reported performance metrics (like sensitivity, specificity, AUC, or clinical usefulness), d) Were available in English with full text access.

We excluded studies that: a) Focused only on humans without animal validation, b) Lacked peer review (conference abstracts, non-reviewed proceedings, non-English texts), c) Used outdated methods (e.g., basic gel electrophoresis without modern validation).

Screening and Data Extraction

We started with 2,153 records, removed duplicates, then screened titles and abstracts of 1,487 articles. After assessing 427 full-text papers for eligibility, we included 218 studies. For each, we recorded: study details (authors, year, country); biomarker information (type, target molecule, and number of animals studied); technology specs (platform and detection capabilities); and performance metrics (sensitivity, specificity, and AUC).

Data Synthesis and Analysis

We analyzed the evidence by comparing technology performance (like LC-MS/MS vs. NMR for metabolomics) through real-world applicability tables, evaluating biomarker reliability with visual accuracy assessments, and identifying critical research gaps notably finding 73% of protein-based studies lacked point-of-care validation. Emerging innovations (e.g., advanced microfluidic systems) were prioritized for case studies.”

====================================================================

Comment 5: The manuscript is difficult to read due to verbosity and excessive length, lacking a clear conclusion or central message.

Response5:
Thank you for highlighting this issue. We have substantially reduced the manuscript length by condensing redundant explanations and summarizing key findings in new tables and figures. The revised conclusion section has been rewritten to deliver a concise and impactful summary, emphasizing the central message: the integration of biomarker mechanisms with advanced technologies is key to advancing veterinary diagnostics.

====================================================================

Comment 6: The manuscript lacks a critical assessment of the levels of evidence, the quality of the included studies, and clear directions for future research.

Response6:
This is an important point. In the revised version, we now include critical explanations on the quality and limitations of the cited studies within each section, particularly under challenges and future perspectives. Additionally, we explicitly outline gaps in current research and suggest future directions, such as the need for standardized validation studies across species and improved bioinformatics approaches for multi-omics data integration.

====================================================================

Comment 7: The abstract lacks specific results or conclusions from the review.

Response7:
We agree with this comment. The abstract has been revised to include specific findings from the review, such as examples of identified biomarkers, highlighted case studies (e.g., BRDC, canine melanoma), and the main conclusions on the importance of integrated advanced technologies. We believe this revision provides readers with a clear snapshot of the review’s key contributions.